# AXIN1 in Plasma or Serum Is a Potential New Biomarker for Endometriosis

**DOI:** 10.3390/ijms20010189

**Published:** 2019-01-07

**Authors:** Malin Ek, Bodil Roth, Gunnar Engström, Bodil Ohlsson

**Affiliations:** 1Department of Internal Medicine, Skåne University Hospital, Lund University, 221 00 Lund, Sweden; malin.ek@med.lu.se (M.E.); bodil.roth@med.lu.se (B.R.); 2Department of Clinical Sciences in Malmö, Clinical Research Centre, Lund University, Box 50332, 202 13 Malmö, Sweden; gunnar.engstrom@med.lu.se

**Keywords:** AXIN1, endometriosis, inflammatory profile, ST1A1, gastrointestinal symptoms

## Abstract

Although endometriosis is considered an inflammatory disease, no reliable diagnostic biomarkers exist for use in clinical practice. The aim was to investigate the inflammatory profile in endometriosis using an exploratory approach of inflammation-related proteins. Patients with laparoscopy-verified endometriosis (*N* = 172), women with microscopic colitis (*N* = 50), healthy controls (*N* = 31), and age-matched controls from the general population (*N* = 100) were enrolled and questionnaires regarding socioeconomic factors, lifestyle habits, and medical history were completed. Sera from patients and healthy controls were analyzed for 92 inflammatory biomarkers using Proximity Extension Assay technology (PEA). Plasma AXIN1 levels were analyzed in patients with endometriosis and controls from the general population by ELISA. General linear model adjusted for age, Mann–Whitney *U*-test, and principal component analysis (PCA) were used for statistical calculations. Serum levels of AXIN1 and ST1A1 were increased in endometriosis compared with MC (*p* < 0.001) and healthy controls (*p* = 0.001), whereas CXCL9 levels were decreased. Plasma levels of AXIN1 were elevated in endometriosis compared with age-matched controls from the general population (30.0 (17.0–38.0) pg/mL vs. 19.5 (15.0–28.0) pg/mL, *p* < 0.001). PCA analysis identified four clusters of proteins, where one cluster differed between endometriosis and controls, with strong correlations for AXIN1 and ST1A1. Plasma/serum AXIN1 is an interesting biomarker to be further evaluated in endometriosis.

## 1. Introduction

Endometriosis is a benign gynecological disease affecting 6–10% of reproductive-age women [1]. The considerable diagnostic delay in the disease [2,3], is partly explained by surgery being the gold standard for diagnosis [1]. No reliable laboratory biomarkers for the disease exist at present, which could contribute to reduced diagnostic delay [3,4].

The most accepted theory of pathogenesis to endometriosis is Sampson’s theory of retrograde menstruation. However, up to 90% of healthy women undergo retrograde menstruation [5], which implies that other mechanisms, such as immunological, genetic, and environmental factors are involved. Endometriosis is widely viewed as an inflammatory condition, and activated macrophages and cytokines such as interleukins 6 (IL-6), 8 (IL-8), and 1β (IL-1β), tumor necrosis factor alpha (TNF-α), and macrophage migration inhibitory factor (MIF) are increased in peritoneal fluid of women with endometriosis [1]. Also, several inflammatory biomarkers are elevated in serum including C-reactive protein (CRP), IL-4, IL-6, IL-8, TNF-α, monocyte chemoattractant protein-1 (MCP-1), RANTES/C-C motif chemokine 5 (CCL5), and YKL-40 [3]. A defective immune system with inadequate responses to clear endometrial fragments has been proposed to be of pathophysiologic importance [6].

The mechanisms behind the gastrointestinal (GI) symptoms in endometriosis have not yet been clarified. Hypotheses regarding these symptoms include altered bowel function due to pelvic inflammatory activity or bowel infiltration of endometriosis lesions [7,8]. Third, GI symptoms in endometriosis could be due to comorbidity with irritable bowel syndrome (IBS) [8], since there is evidence of a visceral hypersensitivity in both endometriosis [9] and IBS [10].

The primary aim of the present study was to investigate the serum inflammatory profile in endometriosis using an exploratory approach of 92 inflammation-related proteins, to enable both validation of previous findings and discovery of new potential inflammatory proteins not yet examined in relation to endometriosis. Secondary aims were to investigate associations of inflammation-related proteins and GI symptoms and disease characteristics in endometriosis.

## 2. Results

### 2.1. Basal Characteristics

Patients were included in two time periods. In the first inclusion period, 307 women who fulfilled inclusion criteria were identified. Of those, 145 women declined to participate, 49 women had moved from the region, and four women denied the diagnosis. Furthermore, nine women were excluded because of an uncertain diagnosis, leaving 100 women included with a mean age of 37.1 ± 7.4 years. The most common localization of endometriosis was isolated ovarian lesions (38%), while 55% had lesions in other anatomical parts of the pelvic cavity (with or without ovarian endometriosis), 4% in other localizations, and 3% had lesions in unspecified localizations. Current hormonal treatment in the patients included combined oral contraceptives (24%), progestogens (23%), gonadotropin-releasing hormone (GnRH) analogs (9%), and estrogen (2%). The prevalence of IBS-like symptoms was 42%.

Patients with endometriosis were significantly younger compared to both women with microscopic colitis (MC) and healthy controls (*p* < 0.001). Women with MC were more likely to smoke compared to women with endometriosis (*p* < 0.001). Body mass index (BMI) did not differ between groups (Table 1).

In a second inclusion period, 266 women fulfilling the inclusion criteria were identified. Of those, 162 women declined to participate, 23 women had moved from the region, and nine women had an uncertain diagnosis, leaving 72 women included. There was no difference in age between the total number of women with endometriosis (*N* = 172) and controls from the general population (*N* = 100) (37.3 ± 7.3 years vs. 36.2 ± 8.1 years, *p* = 0.302). In the total endometriosis cohort, current hormonal treatment included combined oral contraceptives (23.3%), progestogens (17.4%), GnRH analogs (8.7%), and estrogen (1.2%). The prevalence of IBS-like symptoms in endometriosis was 42.4% and 17.0% of controls suffered from IBS.

### 2.2. Protein Expression

From the first included endometriosis cohort, 98 women were selected for the proximity extension assay (PEA) analysis, along with 50 women with MC and 31 healthy controls. Four women with endometriosis, three women with MC, and three healthy controls were excluded due to technical problems (Table 1). Of the inflammation-related proteins (Appendix A), 22 were excluded because of >20% of values below limitation of detection (LOD) in both patients and controls, and one protein was excluded because of technical difficulties (Appendix A). The other 69 proteins were included in the analyses (Appendix A).

AXIN1 and sulfotransferase 1A1 (ST1A1) were the two most significantly elevated proteins in endometriosis compared to MC and controls (Table 2) (Figure 1). Receiver operating characteristic (ROC) curves showed high area under the curves (AUCs) for these proteins, 0.830 and 0.839, respectively, indicating high sensitivity and specificity for endometriosis (*p* < 0.001) (Figure 2).

In the enzyme-linked immune-sorbent (ELISA) analysis of AXIN1 from all 172 endometriosis patients, AXIN1 levels were higher in patients than in age-matched controls from the general population (30.0 (17.0–38.0) pg/mL vs. 19.5 (15.0–28.0) pg/mL, *p* < 0.001).

C-X-C motif chemokine 9 (CXCL9) was the protein with the most significant decrease compared to both MC and healthy controls (*p* < 0.001) (Table 2) (Figure 1).

When comparing protein expression between MC and healthy controls, there was no significant difference between the groups. However, tumor necrosis factor superfamily member 14 (TNFSF14), cluster of differentiation 40 (CD40), hepatocyte growth factor (HGF) and CXCL9 had a tendency towards slightly higher values in MC compared to healthy controls (*p* = 0.012, *p* = 0.017, *p* = 0.024 and *p* = 0.043, respectively) (Table 2).

### 2.3. Principal Component Analysis of Protein Expression

In a principal component analysis (PCA), four components explained 44% (factor 1), 9.8% (factor 2), 4.7% (factor 3), and 3.8% (factor 4), respectively, of the total variance. A compilation of the proteins and their correlations with the four factors is available in Appendix A. Factor 2 differed significantly in endometriosis compared to MC (*p* < 0.001) and healthy controls (*p* = 0.001), when adjusted for age (Table 3). The proteins with the highest correlation within factor 2 were AXIN1 (*r* = 0.886), sirtuin 2 (SIRT2) (*r* = 0.869), stam binding protein (STAMBP) (*r* = 0.863), and ST1A1 (*r* = 0.821) (Appendix A). 

### 2.4. Gastrointestinal Symptoms and Protein Expression

Oncostatin-M (OSM) was the only protein which correlated with several GI symptoms. The strongest correlations regarding symptoms were found between constipation and OSM, monocyte chemotactic protein 1 (MCP-1) and tumor necrosis factor receptor superfamily member 9 (TNFRSF9) (Table 4). OSM levels did not differ between endometriosis patients and MC or healthy controls (data not shown). In MC, no correlations between any GI symptoms and protein levels reached statistical significance (Appendix A). The proteins AXIN1, ST1A1, CXCL9, and OSM, which showed the greatest differences between endometriosis and controls or symptoms, were not affected by the presence of IBS-like symptoms, localization of endometriosis lesions or hormonal treatment (data not shown). In alignment, AXIN1 levels were not affected by the presence or absence of IBS in controls from the general population (data not shown).

## 3. Discussion

The major finding was that serum concentrations of several inflammation-associated proteins were altered in women with endometriosis compared to MC and healthy controls. AXIN1 was the protein with the most pronounced increase, followed by ST1A1, whereas CXCL9 was the protein with most significant decrease. The cytokine OSM correlated with several GI symptoms in endometriosis.

To our knowledge, increased serum levels of AXIN1 and ST1A1 in women with endometriosis are novel findings, and need to be confirmed in further studies. AXIN1 was the most interesting protein in the analyses, with a high correlation with factor 2 in the PCA, a factor that separated endometriosis from controls. AXIN1 is a cytoplasmic protein, and a negative regulator of the Wnt signaling pathway by downregulation of β-catenin [11]. Expression of β-catenin is increased in endometriosis lesions compared to normal endometrium [12] and there is evidence to suggest that the Wnt signaling pathway is involved in the pathogenesis of endometriosis [12,13]. It is hypothesized that aberrant activation of the Wnt pathway in endometrium may enable development of endometriosis by increased cell migration and invasion. Targeting of this pathway may be a potential method to treat endometriosis [13]. AXIN1 was not related to GI symptoms or disease characteristics. The increase of AXIN1 in sera/plasma may solely reflect an aberrant activation of the Wnt pathway in endometriosis, and may not reflect causality. Since AXIN1 analyses are not commonly used, we do not know whether other inflammatory and infectious diseases express higher levels of this protein. The combination of clinical suspicion of endometriosis in young women and AXIN1 levels in serum/plasma has to be further evaluated in daily practice, to determine its role in clinical practice. IBS in controls and IBS-like symptoms in patients did not affect AXIN1 levels, which is why the protein could be a potential new biomarker to separate IBS from endometriosis, and thereby reduce the diagnostic delay in endometriosis. However, the ELISA kit needs to be improved with higher sensitivity in the analysis.

ST1A1 catalyzes sulfur conjugation of substances such as drugs, neurotransmitters, bile acids, and hormones, and also plays a significant role in the metabolism of estrogens [14]. Inflammatory activity can affect the activity of ST1A1 [15], which is why it is considered as a potential novel inflammatory biomarker. Since endometriosis is an estrogen-dependent disease, it is interesting that sulfotransferase 1 E1 mRNA levels are increased, and sulfotransferase 2 B1 mRNA levels are decreased, in endometriosis lesions [16]. Since ST1A1 levels did not correlate with GI symptoms or disease characteristics, this increase may merely reflect an increased inflammatory activity in endometriosis. ST1A1 may also indicate aberrant serum estrogen levels and use of hormonal therapy. However, ST1A1 levels were not altered in patients with estrogen treatment, in comparison to untreated patients.

The most studied chemokine in endometriosis is IL-8 (also called CXCL 8), which in a systematic review had the best potential among chemokines as a diagnostic marker for endometriosis [17]. Unfortunately, IL-8 was excluded in the present study due to technical problems. Previous studies have reported significantly lower concentrations of the closely related CXCL9 and CXCL10 (also called IP-10) in peripheral blood in endometriosis [18,19,20], in line with the present study. There is increased expression of MMP-9 in endometrium [21] and in peritoneal fluid in endometriosis [22], and MMP-9 cleaves CXCL9 at three different sites and degrades CXCL10 [23]. It is possible that the increased levels of MMP-9 in endometriosis result in cleavage and degradation of CXCL9 and CXCL10, causing decreased levels of these proteins.

The cytokine OSM correlated with several GI symptoms in endometriosis, most strongly constipation. To our knowledge, the connection between serum OSM-levels and GI symptoms has not been investigated previously. OSM is produced by activated T-lymphocytes and monocytes, and induces increased levels of IL-6 [24], a cytokine elevated in endometriosis [17]. In the present study, neither serum levels of OSM nor IL-6 were elevated in endometriosis. In inflammatory bowel disease (IBD), OSM and its receptor (OSMR) are expressed in high levels in inflamed intestinal tissue, and levels are correlated with histopathological disease severity and CRP levels [25,26]. OSM may hypothetically contribute to the establishment and maintenance of endometriosis lesions through its binding on extracellular matrix (ECM) components in a way that protects from proteolytic degradation, therefore amplifying inflammatory responses and driving disease chronicity [26].

MCP-1 (also called CCL2) regulates migration and infiltration of monocytes/macrophages [27], cells found in increased numbers in the peritoneal cavity in endometriosis [28]. MCP-1 correlated with constipation in endometriosis, and levels of MCP-1 were significantly lower in endometriosis compared to MC, but in the same magnitude as in controls. In a systematic review, 50% of studies reported increased levels and 50% found equal levels in endometriosis compared to controls [17]. Increased serum levels of MCP-1 have also been reported in IBS, independently of subgroups [29,30]. The role of non-coding RNAs has been discussed in relation to endometriosis, and circulating microRNAs have been proposed as potential biomarkers for the disease [31,32].

There are several limitations of this study. The study is of a cross-sectional design and cannot investigate causal relationships. Another disadvantage is that patients with MC and healthy controls differed significantly in age compared to women with endometriosis. To minimize this impact, all comparisons between groups were adjusted for age. However, the fact that some of the women with MC were menopausal may affect the differences in protein levels. We could not find any studies regarding the impact of menopause on the proteins that differed the most. Generally, several cytokines are elevated in postmenopausal women, e.g., IL-1β, IL-6, and IL-8, which is why the actual estimated differences may be underestimated [20,33,34]. The phase of the menstrual cycle was neither known in controls nor in endometriosis, but the majority of endometriosis patients were on continuous hormonal treatment affecting the cycle. Continuous combined oral contraceptives have been shown to significantly reduce pelvic pain and improve quality of life in endometriosis [35,36]. Hormonal treatment may also affect protein levels. Oral hormonal therapy with estrogen, with or without progestogens, has been shown to increase the blood levels of CRP and to reduce other inflammatory factors such as MCP-1 and TNF-α [37]. Current hormonal treatment was not known in controls, which is a limitation. Although no studies have examined the effect of the menstrual phase on AXIN1, ST1A1, and CXCL9 levels, this should be examined in future studies, to exclude impact. However, these proteins that showed the greatest differences between endometriosis and controls were not affected by hormonal treatment in endometriosis, which suggests that hormone levels, either dependent on treatment or menstrual phases, are of minor importance to their serum levels. Protein levels in PEA are expressed as arbitrary units, and cannot be compared to other protein levels. MC is sometimes treated with immunosuppressive medication, most often budesonide which is inactivated by first-passage metabolism in the liver, and does not exert systemic effects with altered levels of inflammatory proteins. However, the vast majority of included patients were in an inactive state of the disease and had no treatment. Furthermore, protein levels did not differ between MC and healthy controls who had no immunosuppressive treatment. The low number of healthy controls may explain the fewer significant differences between endometriosis and controls than between endometriosis and MC.

## 4. Methods

This study was approved by the Ethics Review Board of Lund University, No 2012/564 (09/10/2012), 2016/56 (03/05/2016), 2009/565 (03/11/2009), 2011/209 (16/05/2012) and 2012/594 05/10/2012). All subjects gave their written, informed consent before inclusion in the study.

### 4.1. Patients

Patients with endometriosis were recruited at the Department of Gynecology at Skåne University Hospital in Malmö, Sweden. Patients were identified by the International Statistical Classification of Diseases and Related Health Problems, ICD-10, N 80. The recruitment was conducted between March 2013 and July 2014, and between September 2016 and March 2017. The inclusion criteria were a definite diagnosis of endometriosis confirmed by laparoscopy or laparotomy and an ability to comprehend the Swedish or English language. Exclusion criteria were severe or multiple comorbidities of somatic diseases (e.g., cancer, multiple sclerosis, or several severe concomitant somatic diseases) and severe mental illness (severe substance abuse or psychotic disease), a diagnosis of IBD, living too far from the geographical area of the hospital, an uncertain endometriosis diagnosis (a diagnosis confirmed with ultrasonography or a clinical examination) or current pregnancy.

### 4.2. Controls

Women who had been diagnosed with MC, K52.8, at the Department of Gastroenterology have retrospectively been enrolled in a previous study [38]. These women served as comparison, since MC is characterized by a low-grade intestinal inflammation. Women with persistent GI symptoms and histopathological changes were classified as MC, but women who only had had one incidence of GI symptoms and histopathological changes, and denied actual symptoms of MC or IBS and had no present signs of disease, were classified as having suffered from a transient episode of MC, and served as healthy controls. In total, 50 women with persistent MC and 31 healthy controls were identified, where both blood samples and complete health declarations were available.

The Malmö Offspring Study (MOS) consists of offspring to subjects who participated in the Malmö Diet and Cancer cardiovascular cohort (MDCS; *N* = 6103) [39]. Recruitment of participants is currently ongoing. A randomly selected cohort from MOS was previously used to study basal characteristics and GI symptoms in the population, where the study is described in more detail [40]. From this cohort, 100 age-matched women served as controls from the general population in the present study.

### 4.3. Study Design

The present study is of a cross-sectional character. The patients who agreed to participate were interviewed, and both patients and controls completed questionnaires regarding socioeconomic factors, lifestyle habits, and medical history. They also completed the Visual Analogue Scale for Irritable Bowel Syndrome (VAS-IBS), a validated questionnaire addressing common GI symptoms. Blood samples were drawn and plasma and sera were kept frozen at −80 °C. Patients and healthy controls were analyzed for 92 inflammatory biomarkers using a PEA technology. AXIN1 levels were analyzed in patients with endometriosis and controls from the general population by ELISA.

### 4.4. The Visual Analogue Scale for Irritable Bowel Syndrome

The VAS-IBS is a validated questionnaire estimating the GI symptoms of abdominal pain, diarrhea, constipation, bloating and flatulence, vomiting and nausea, psychological well-being, and intestinal symptom’s influence on daily life. The items are measured on a scale from 0 to 100, where 100 represents very severe symptoms and 0 lack of symptoms. The values are inverted from the original VAS-IBS scale. Two additional items, if the subject experiences defecation urgency or a sensation of incomplete evacuation when defecating, are answered with yes or no [41].

### 4.5. Analytical Methods

Inflammatory proteins in serum were analyzed using a multiplex PEA, Proseek® Multiplex Inflammation 1 kit (Olink Bioscience, Uppsala, Sweden), in which 92 proteins are simultaneously analyzed (Appendix A) [42,43]. The analyses were conducted at the Clinical Biomarkers Facility, Science for Life Laboratory, Uppsala, Sweden, according to the manufacturer’s instructions. Briefly, oligonucleotide-labeled antibody pairs bind to their respective target protein in the sample. A polymerase chain reaction (PCR) reporter sequence is formed by proximity-dependent DNA polymerization, and is detected and quantified using real time PCR. Information regarding data validation, LOD, specificity, and reproducibility is available via Olink’s website [44]. Data are expressed as Normalized Protein Expression (NPX) values. The NPX unit is an arbitrary value on a log2 scale, where a larger NPX value represents a higher protein expression in the sample. Human AXIN1 was analyzed with sandwich ELISA technology (MBS762601, MyBiosource, San Diego, CA, USA, lot.no.H2497C119) according to the manufacturer’s manual. Standards (78–5000 pg/mL) and human EDTA plasma, undiluted or diluted 1:2 in dilution buffer, were incubated in a plate pre-coated with an anti-AXIN1 antibody. After incubation and washing procedure, biotin-labelled AXIN1 antibodies were added to the wells. The unbound antibodies were washed off and HRP-streptavidin conjugate was added. Incubation and wash procedure were replicated as before, and tetramethylbenzidine (TMB) substrate was added to form a horseradish peroxidase (HRP) enzymatic reaction. The reaction was stopped with an acidic stop solution and the absorbance was measured at 450 nm. The assay recovery range for plasma (*N* = 5) was 86%–99 %; intra- and inter-assay coefficients of variance (CV) were both ≤8%. Sensitivity of analysis was <94 pg/mL. Due to the low concentration in plasma and high background, values from patients and controls were under the lowest standard (78 pg/mL) and outside the measuring range, which is why values had to be extrapolated from below the standard curve.

### 4.6. Statistical Methods

Statistical calculations were performed using SPSS© for Windows (release 23.0; IBM). Proteins with >20 % of values below LOD in both patients and controls were excluded (Appendix A). Descriptive statistics were calculated using Student’s *t*-test or Mann–Whitney *U*-test, depending on the normality distribution of data. Dichotomous variables were analyzed by Fischer’s exact-test. Since age differed between patients and controls, group comparisons of protein levels were conducted using a general linear regression model, where age was added as a covariate. If protein levels differed between patients and controls, BMI and smoking were added as covariates. These factors did not have any impact on the outcome of the statistical tests and were therefore excluded in the statistical models. ROC-curves with an AUC and a 95% confidence interval (CI) were calculated for AXIN1 and ST1A1. AXIN1 levels from ELISA were compared by the Mann–Whitney *U*-test. PCA was conducted to identify uncorrelated factors among the various plasma proteins. The scree-test was used to determine which factors to retain. In the next step, these factors were compared between patients and controls using a general linear regression model, where age was added as a covariate. BMI and smoking were not significantly associated with any of the four factors and these results were therefore not shown. Spearman’s correlation test was used for correlations between protein levels and GI symptoms. Values are presented as numbers and percentages, mean ± standard deviation (SD) or 95% CI and median (interquartile range). A *p*-value ≤ 0.01 was considered statistically significant, to adjust for multiple comparisons.

## 5. Conclusions

This exploratory study provides new knowledge of the systemic inflammatory response in endometriosis. We conclude that the serum levels of several inflammation-related proteins were altered in endometriosis. The most increased proteins were AXIN1 and ST1A1, while CXCL9 was the most decreased protein. Protein levels were not associated with anatomical localization or hormonal treatment of endometriosis lesions and showed sparse correlations with GI symptoms. AXIN1 seems to be the most promising protein to be further evaluated in clinical practice, but the ELISA kit has to be improved before being more widely used. Additionally, future analyses related to the phase of the menstrual cycle in other cohorts are necessary to confirm these findings. OSM levels correlated with several GI symptoms in endometriosis.

## Figures and Tables

**Figure 1 ijms-20-00189-f001:**
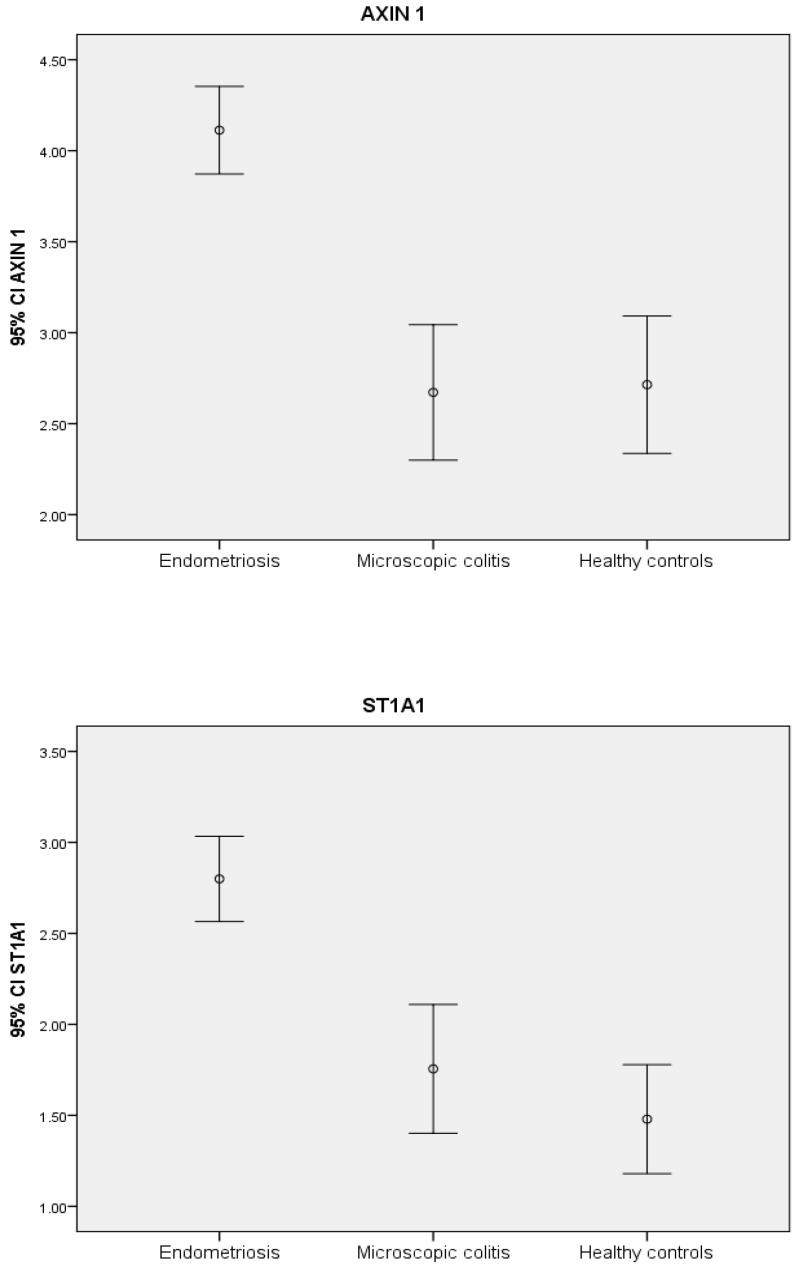
NPX levels in endometriosis and controls. NPX = normalized protein expression. The NPX unit is an arbitrary unit of concentrations. Figures indicate mean NPX value and 95% confidence interval of the mean value in each group; endometriosis (*N* = 94), microscopic colitis (*N* = 47) and healthy controls (*N* = 28).

**Figure 2 ijms-20-00189-f002:**
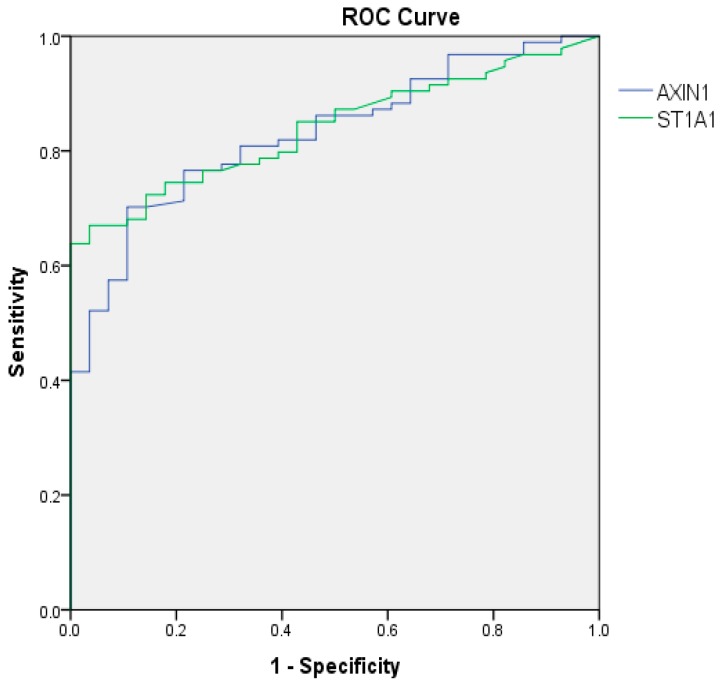
Receiver operating characteristic (ROC) curves of the most significantly elevated expressed proteins in endometriosis compared to controls. ROC = receiver operating characteristics, ST1A1 = sulfotransferase 1A1. ROC curves are calculated by comparing endometriosis patients (*N* = 94) to healthy controls (*N* = 28). The area under curve (AUC) for AXIN 1 was 0.830 (95% confidence interval (CI): 0.755–0.904), *p* < 0.001 and AUC for ST1A1 was 0.839 (95% CI: 0.771–0.908), *p* < 0.001.

**Table 1 ijms-20-00189-t001:** Patient and control group characteristics in endometriosis and controls included in the proximity extension assay analysis.

Characteristics	Endo	MC	HC	*p*-Value	*p*-Value
	*N* = 94	*N* = 47	*N* = 28	Endo vs. MC	Endo vs. HC
Age (years)	36.7 ± 7.4	59.4 ± 9.9	64.1 ± 5.5	<0.001	<0.001
BMI (kg/m^2^)	24.0 (21.8–26.2)	23.9 (21.8–27.8)	24.1 (20.8–25.7)	0.718	0.490
Smoking (*n*, %)	14 (14.9)	24 (51.1)	6 (21.4)	<0.001	0.562

Endo = endometriosis, MC = microscopic colitis, HC = healthy controls, BMI = body mass index. Values are expressed as mean ± standard deviation, median (interquartile range) and number, %. Student’s *t*-test, Mann–Whitney *U*-test, and Fischer’s exact test. *p* ≤ 0.05 was considered statistically significant.

**Table 2 ijms-20-00189-t002:** Protein expression in endometriosis and controls.

Protein	Absolute Value	Absolute Value	Absolute Value	*p*-Value	*p*-Value
	Endometriosis	MC	HC	Endometriosis vs. MC	Endometriosis vs. HC
	*N* = 94	*N* = 47	*N* = 28		
AXIN1	4.1 ± 1.2	2.7 ± 1.3	2.7 ± 1.0	<0.001	0.001
ST1A1	2.8 ± 1.1	1.8 ± 1.2	1.5 ± 0.8	0.001	0.010
CXCL9	7.1 ± 0.7	8.7 ± 0.9	8.4 ± 0.7	<0.001	<0.001
CD40	9.4 ± 0.6	9.2 ± 0.4	9.1 ± 0.4	0.007	0.020
SIRT2	6.2 ± 1.4	4.9 ± 1.2	4.9 ± 1.1	0.001	0.018
CD244	6.0 ± 0.4	5.8 ± 0.4	5.8 ± 0.4	0.002	0.131
STAMBP	4.6 ± 1.2	3.7 ± 0.9	3.7 ± 1.0	0.001	0.041
NT3	1.9 ± 0.5	1.4 ± 0.3	1.6 ± 0.6	0.001	0.226
MCP-1	8.6 ± 0.4	9.2 ± 0.6	9.1 ± 0.3	0.003	0.021
MMP-10	8.6 ± 0.7	9.5 ± 0.9	9.2 ± 0.6	0.010	0.335
CCL11	7.3 ± 0.4	8.3 ± 0.5	8.3 ± 0.5	<0.001	0.030
TGF-α	0.8 ± 0.3	1.0 ± 0.2	0.9 ± 0.3	<0.001	0.016
OPG	9.7 ±0.4	10.1 ± 0.5	10.0 ± 0.3	0.015	0.004
CDCP1	2.0 ± 0.4	2.9 ± 0.7	3.0 ± 0.5	0.047	0.008
Flt3l	8.7 ± 0.4	9.2 ± 0.5	9.1 ± 0.5	0.023	0.055
CXCL10	8.4 ± 0.9	9.3 ± 0.8	9.6 ± 1.0	0.041	0.015
IL-18R1	7.2 ± 0.5	7.3 ± 0.6	7.3 ± 0.5	0.437	0.038
HGF	6.6 ± 0.4	7.1 ± 0.4	6.9 ± 0.4	0.017	0.740
TNFRSF9	5.8 ± 0.4	6.3 ± 0.6	6.1 ± 0.5	0.013	0.338
IL-10	3.5 ± 0.6	4.0 ± 0.6	3.8 ± 0.7	0.036	0.702
IL-7	3.1 ± 0.8	3.1 ± 0.7	3.0 ± 0.7	0.047	0.047
TNFSF14	2.5 ± 0.6	2.6 ± 0.4	2.3 ± 0.4	0.819	0.038
EN-RAGE	2.3 ± 1.1	1.9 ± 0.8	1.8 ± 0.5	0.017	0.013

HC = healthy controls, MC = microscopic colitis. Values are expressed as mean ± standard deviation. Data are expressed as Normalized Protein Expression (NPX) values. The NPX unit is an arbitrary value on a log2 scale, where a larger NPX value represents a higher protein expression. Comparisons were calculated using a general linear regression model with age as a covariate. Proteins with *p*-values < 0.05 in any comparison are shown. *p* ≤ 0.01 was considered statistically significant.

**Table 3 ijms-20-00189-t003:** Factor analysis of inflammatory biomarkers in endometriosis compared to controls.

PCA Factors	Endo	MC	HC	*p*-Value	*p*-Value
	*N* = 94	*N* = 47	*N* = 28	Endo vs. MC	Endo vs. HC
Factor 1	−0.14 ± 0.61	0.35 ± 0.57	0.25 ± 0.49	0.381	0.777
Factor 2	0.52 ± 0.92	−0.63 ± 0.73	−0.69 ± 0.57	<0.001	0.001
Factor 3	−0.45 ± 0.72	0.60 ± 0.78	0.31 ± 0.66	0.068	0.966
Factor 4	0.27 ± 0.84	−0.39 ± 1.11	−0.31 ± 0.99	0.372	0.931

Endo = endometriosis, HC = healthy controls, MC = microscopic colitis. *p*-values were calculated using a general linear regression model where age was added as a covariate. Values are expressed as mean ± standard deviation of the sum of protein levels in each factor. *p* ≤ 0.01 was considered statistically significant.

**Table 4 ijms-20-00189-t004:** Correlations of gastrointestinal symptoms and protein levels in endometriosis.

Protein	Abdominal Pain	Diarrhea	Constipation	Bloating and Flatulence	Vomiting and Nausea
OSM	*r* = 0.182, *p* = 0.083	*r* = 0.235, *p* = 0.024	*r* = 0.268, *p* = 0.010	*r* = 0.238, *p* = 0.022	
MCP1			*r* = 0.291, *p* = 0.005		
TNFRSF9			*r* = 0.280, *p* = 0.007		
CCL 11			*r* = 0.215, *p* = 0.039		
MMP10		*r* = -0.229, *p* = 0.028			
EN-RAGE		*r* = -0.228, *p* = 0.029			
IL 5			*r* = 0.237, *p* = 0.023		

CCL 11 = eotaxin-1, EN-RAGE = protein S100-A12, IL 5 = interleukin 5, OSM = oncostatin-M, MCP 1 = monocyte chemotactic protein 1, MMP10 = matrix-metalloproteinase-10, TNFRSF9 = tumor necrosis factor receptor superfamily member 9. Correlations were calculated using Spearman’s correlation test. *p*-values < 0.1 are presented. *p*-values ≤ 0.01 were considered statistically significant.

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
