# Peer review of "AXIN1 in Plasma or Serum Is a Potential New Biomarker for Endometriosis"

_ijms, 2019, doi:10.3390/ijms20010189_

Reviewer 2 Report

This is an exploratory cross sectional study using using three different cohorts of women to compare inflammatory proteins in endometriosis.  The study is well presented and statistical methods are appropriate. However, even when the scientists controlled for age, it is still possible that the differences in proteins could be related to the women with endometriosis being premenopausal and the other cohorts being women already in menopause.  It would be necessary and relevant to add to the discussion a clarification about how certain inflammatory proteins change during menopause and in response to estrogen and progesterone.  On this same line, did researchers recorded whether the women with endometriosis were in proliferative vs. secretory phase at the time of sample collection?  This would be extrenmely important to analyze compared to the other groups. Withouth this analysis, it is hard to decide the feasibility of future studies in AXIN1. 

     Author Response

     Thank you very much for your valuable criticism of our manuscript. We have now tried to answer all your questions, and to revise the manuscript according to your suggestions. All changes are marked in yellow. We think that the manuscript has been improved after revision and language edition. Please reconsider this manuscript for publication in International Journal of Molecular Sciences.

     This is an exploratory cross sectional study using three different cohorts of women to compare inflammatory proteins in endometriosis.  The study is well presented and statistical methods are appropriate. However, even when the scientists controlled for age, it is still possible that the differences in proteins could be related to the women with endometriosis being premenopausal and the other cohorts being women already in menopause.  It would be necessary and relevant to add to the discussion a clarification about how certain inflammatory proteins change during menopause and in response to estrogen and progesterone.

     Response: None of the endometriosis patients were in menopause, and none of the women in the age-matched control group for the ELISA analysis of AXIN1 were in menopause or postmenopausal. However, it is likely that some of the women in the control group or microscopic colitis (MC) for the PEA analysis were in menopause. We could not find any studies concerning the proteins that differed the most between endometriosis and controls, regarding hormonal changes in the levels in the menopause. Generally, studies have shown that postmenopausal women have elevated levels of several serum inflammatory cytokines, e.g. IL-1β, IL-6, IL-8 and TNF-α. A few cytokines have been shown to be decreased compared to fertile women, e.g. IL-20 (Malutan AM et al 2014, Cioffi M et al. 2002, Kim OY et al. 2012).  It is therefore possible that the differences in protein levels between patients and controls are underestimated. The possible impact of menopause on protein levels in the control group and MC is a limitation of the study, which is now discussed in the discussion section, page 4, line 185-189.

Several inflammatory proteins are associated with hormonal therapy. For instance, C-reactive protein is elevated, and MCP-1 and TNF-α are decreased, in women treated with estrogen, alone or in combination with progesterone (Koh KK et al. 2006), which is now added to the discussion, page 5, line 193-195. We could not find any studies regarding impact of hormonal levels on levels of ST1A1 specifically in human; only rat studies are performed. Regarding AXIN1, in vitro studies of breast cancer cells have shown estrogen-mediated AXIN1 suppression (Chimge et al Nature Communications 2016). No studies have examined circulating AXIN1 levels in relation to hormonal treatment or menstrual phases. In endometriosis, hormonal treatment was not associated with protein levels, page 3, line 115-118. However, we had no data in hormonal treatment in controls, which is a drawback of the study and is now added as a limitation, page 5, line 195-196.

    On this same line, did researchers recorded whether the women with endometriosis were in proliferative vs. secretory phase at the time of sample collection?  This would be extrenmely important to analyze compared to the other groups.  Without this analysis, it is hard to decide the feasibility of future studies in AXIN1.

     Response: A limitation of the study is that we did not have information regarding whether the endometriosis patients or controls were in proliferative or secretory phase at the time of sample collection. However, a majority of patients (59%) had a continuous current hormonal treatment, page 2, line 62-65 and 74-76, why the menstrual phases are suppressed and cannot be determined. Most had continuous treatment so they did not have any menstruation at all. But even if they have combined p-pills with menstruation, the cycle is not the same as usual. We do not have the menstrual phase of the controls either. However, since the controls in the MOS cohort is collected from hundreds of persons from the general population, the phase of menstrual cycle should be randomly present. Also, 172 women with endometriosis should be randomly distributed in the menstrual phase, if menstruation is present. We can not find a single study examining the levels of AXIN1 in relation to the menstrual phase. This study has to be regarded as a first pilote study, but further studies are needed to confirm these results in other cohorts, and the phase of the menstrual cycle should be recorded in future studies. However, since hormonal treatment did not affect AXIN1 (page 3, line 115-118), different phases should not either affect AXIN1 levels. Because hormonal treatment does not only reflect addition of hormones, but also an affected menstrual cycle. This is now added in the discussion, page 4-5 line 189-200 and conclusion, page 5, line 216-217. We still think our findings are of significance, and they can further be analyzed in other cohorts in relation to menstruation phases. The role of the study as a pilote trial is stressed on several pages, e.g. discussion page 3, line 127 and page 3, line 139-141.

References

Chimge N-O, Little GH, Baniwal SK, et al. RunX1 prevents oestrogen-mediated AXIN1 suppression and β-catenin activation in ER-positive breast cancer. Nature Comm 2016;7:10751.

Cioffi M, Esposito K, Vietri MT, Gazzerro P, D'Auria A, Ardovino I, Puca GA, Molinari AM. Cytokine pattern in postmenopause. Maturitas. 2002 Mar 25;41(3):187-92.

Kim OY, Chae JS, Paik JK, Seo HS, Jang Y, Cavaillon JM, Lee JH. Effects of aging and menopause on serum interleukin-6 levels and peripheral blood mononuclear cell cytokine production in healthy nonobese women. Age (Dordr). 2012 Apr;34(2):415-25. doi: 10.1007/s11357-011-9244-2. Epub 2011 Apr 13.

Koh, KK; Yoon, BK. Controversies regarding hormone therapy: Insights from inflammation and hemostasis. Cardiovasc Res 2006, 1;70(1):22-30. doi: 10.1016/j.cardiores.2005.12.004

Malutan AM, Dan M, Nicolae C, Carmen M. Proinflammatory and anti-inflammatory cytokine changes related to menopause. Prz Menopauzalny. 2014 Jun;13(3):162-8. doi: 10.5114/pm.2014.43818. Epub 2014 Jun 30.

Round 2

Reviewer 2 Report

Thank you for the clarifications on the manuscript regarding the menopausal stage of the women and their cycle data.

Author Response

Thank you very much for your comments. We do think they improved the manuscript.

Sincerely Bodil Ohlsson